# Steganalysis of Adaptive Multi-Rate Speech Based on Extreme Gradient Boosting

**Congcong Sun [1,2,3], Hui Tian [1,2,3,\*] , Chin-Chen Chang [4,5], Yewang Chen [1,3], Yiqiao Cai [1,3], Yongqian Du [1,3], Yong-Hong Chen [1,3] and Chih Cheng Chen [6,\*]**

1   College of Computer Science and Technology, National Huaqiao University, Xiamen 361021, China; ccsun@hqu.edu.cn (C.S.); Ywchen@hqu.edu.cn (Y.C.); caiyq@hqu.edu.cn (Y.C.); yqdu@hqu.edu.cn (Y.D.); iamcyh@hqu.edu.cn (Y.-H.C.)
2   State Key Laboratory of Information Security, Institute of Information Engineering, Chinese Academy of Sciences, Beijing 100093, China
3   Fujian Key Laboratory of Big Data Intelligence and Security, National Huaqiao University, Xiamen 361021, China
4   Department of Information and Computer Science, Feng Chia University, Taichung 40724, Taiwan; alan3c@gmail.com
5   School of Computer Science and Technology, Hangzhou Dianzi University, Hangzhou 310018, China
6   Information and Engineering College, Jimei University, Fujian 361021, China
\*   Correspondence: htian@hqu.edu.cn (H.T.); 201761000018@jmu.edu.cn (C.C.C.); Tel.: +86-592-6162497 (H.T.)



**Featured Application: The presented scheme can be used to detect the potential steganography in the Voice-over-IP (VoIP) streams, and thereby prevent cybercrimes based on covert VoIP communications.**

**Abstract:** Steganalysis of adaptive multi-rate (AMR) speech is a hot topic for controlling cybercrimes grounded in steganography in related speech streams. In this paper, we first present a novel AMR steganalysis model, which utilizes extreme gradient boosting (XGBoost) as the classifier, instead of support vector machines (SVM) adopted in the previous schemes. Compared with the SVM-based model, this new model can facilitate the excavation of potential information from the high-dimensional features and can avoid overfitting. Moreover, to further strengthen the preceding features based on the statistical characteristics of pulse pairs, we present the convergence feature based on the Markov chain to reflect the global characterization of pulse pairs, which is essentially the final state of the Markov transition matrix. Combining the convergence feature with the preceding features, we propose an XGBoost-based steganalysis scheme for AMR speech streams. Finally, we conducted a series of experiments to assess our presented scheme and compared it with previous schemes. The experimental results demonstrate that the proposed scheme is feasible, and can provide better performance in terms of detecting the existing steganography methods based on AMR speech streams.

**Keywords:** steganalysis; steganography; convergence feature; extreme gradient trees; adaptive multi-rate speech

## 1. Introduction

Steganography is a security technique of embedding secret information into a certain carrier while the secret information can be extracted accurately. Thus, this technology can be applied to covert communication. Such ancient security technology has existed for more than two thousand years [1]. In ancient battlefields, soldiers made use of the board to conceal information to mislead enemies; in a letter, the special shape of some characters was changed and some strokes were added for the same

purpose. Over time, these carriers have evolved from the image [2,3] to other fields, such as text [4,5], then video [6,7], network protocol [8,9], audio [10–12], and Voice over IP (VoIP) [13–16]. In contrast with other carriers, VoIP has many advantages, such as high convert bandwidth, flexible conversation length, and a large amount of carrier data. Therefore, many researchers have been taking VoIP as carriers to embed secret information.

As internet communication technologies (3G and 4G speech services [17–19]) that are mainly used for VoIP develop, we are exposed to digital audiovisual products, including Snapchat, WeChat, and others. In 4G speech services, adaptive multi-rate (AMR) codec is considered to be a significant compression measurement for coding. Furthermore, it is also leveraged to store AMR-encoded spoken audio for a file format. Thanks to its wide application in the communication domain, a large number of researchers have been regarding AMR as ideal carriers for information embedding [14–16,20,21].

AMR speech codec adopts algebraic code-excited linear prediction. Functionally, it consists of three parts, one of which includes algebraic codebook indices (ACIs) also called fixed-codebook (FCB). Lots of studies have been devoted to FCB [20,21], as FCB accounts for a large proportion of AMR speech codecs. Geiser and Vary [20] embedded two secret bits into a track pulse with the modified fixed-codebook-search algorithm. Moreover, to make the process of embedding information more flexible, and adaptive suboptimal pulse combination method was proposed by Miao et al. [21]. To be specific, the steganographic capacity [21] can be regulated through the introduction of embedding factor $\eta$. Taking the AMR speech codec at 12.2 kbps mode as an example, $\eta$ usually has three setting values (1, 2, 4), which correspond to different steganographic bandwidths. Their experiments demonstrated that, by selecting an optimal $\eta$, this method could achieve a good balance between speech quality and embedding capacity.

However, just like a coin has two sides, steganography may be an effective tool for criminals to commit crimes. Especially, AMR speech streams can pass through various firewalls and network listening devices without barriers. In other words, unauthorized data can be leaked and transported, which will result in an information security problem. For example, it may be utilized to carry out terrorist activities. Therefore, the protective measures called steganalysis are a subject worth our studying [22–30], which is applied to detect whether the AMR speech has been inserted into secret messages. In this research, we mainly focus on the steganalysis on AMR speech streams, which has a wide application in VoIP streams.

To detect the above AMR speech steganography methods, several related AMR speech steganalysis methods have been proposed [31–33]. In their methods, they all adopt statistical features as steganalysis features and support vector machines (SVM) as the classifier. First, Miao et al. [31] used Markov transition probabilities (MTP), which were utilized to capture the link between pulse positions. As for the second method, the related information-entropy methods (conditional entropy, joint entropy) were applied to extract the feature, which could represent the distribution of pulse positions. However, the exchange of the pulse positions in the encoding process makes the above two methods fail to distinguish between the cover sample and steganographic sample effectively. Moreover, Ren et al. [32] put forward the statistical characteristics of the same pulse positions (SPP). Specifically, the probabilities of the same pulse positions were adopted as detecting features, which was aimed at the pulse positions in the same track. Nonetheless, a problem occurs. If secret information could not be inserted into the same positions in the track-pulse, Fast-SPP would be out of work. Therefore, more integral features were put forth [33] for the steganalysis of AMR speech. To address the above problem, the statistical features of pulse pairs (SCPP) were raised and it was composed of three parts. The first was the long-term distribution of speech signals, which adopted probabilities of pulse pairs. The second was short-term invariant characteristic of speech using MTP of pulse pairs. The last was the track-to-track correlation also related to MTP, exploiting joint probability matrices to discover the correlation of pulse pairs on different tracks. Furthermore, Adaboost [34] was adopted for feature selection to reduce the dimension and the dimension of the final feature of the steganalysis feature is 498 at AMR 12.2 kbps.

However, the above steganalysis methods have demerits to some degree. First and foremost, the classifiers employed by the above steganalysis methods are SVM, failing to prevent the classification model from overfitting, when the dimension of feature is high. Furthermore, SVM performs poorly on generalization performance. Last but not least, all features related to the MTP neglect its stationary state [35–37]. It represents the global and stable characterization of the MTP and it is called the convergence feature. Specifically, the MTP will reach a convergence state by multiplying itself several times.

In this work, motivated by the above analysis, we present an AMR-based speech steganalysis based on extreme gradient boosting (XGBoost) [38–43]. Our contributions can be summarized as follows. First, we propose a convergence feature for detecting the FCB-based steganography, which describes the global characteristics of AMR speech streams. Since the state-of-the-art feature SCPP [33] mainly reflects local characteristics of AMR speech streams, the presented convergence feature is a useful complement to SCPP. Second, we propose a new steganalysis feature using both the convergence feature and SCPP. Finally, we present a new steganalysis model for AMR speech, which employs XGBoost rather than traditional SVMs employed in previous schemes as the classifier. It enjoys some advantages, such as mining the potential information from the hybrid features, avoiding overfitting (making assumptions too strict to get consistent ones) and having a strong generalization ability.

The remaining of this paper is organized as follows. In Section 2, preliminary knowledge is introduced, consisting of the AMR-based steganography method and a review of the preceding feature for AMR-based steganalysis, then a brief introduction about XGBoost is made including the boost and decision trees. In Section 3, the convergence features grounded in the Markov chain and the proposed scheme based on XGBoost is presented. In Section 4, our presented scheme is evaluated with a set of experiments and the experimental results are analyzed. In the final section, a conclusion about our work is presented.

## 2. Preliminary and Relate Work

To make readers understand our idea, First, we give a brief introduction to the AMR speech steganography [20,21]; we then review the preceding scheme for AMR-based steganalysis; finally, we introduce the XGBoost classification model, which includes boosting and decision trees.

### 2.1. AMR-Based Steganography Method

According to the searching process of the fixed codebook [18], the codebook vector is acquired by the depth-first-tree-search, which is suboptimal. In other words, as for all significant positions, the searching process only accounts for a small proportion. Motivated by this, taking the AMR speech codec at 12.2 kbps, for example, a steganographic method was First presented by Geiser and Vary [20], by restricting the searching range of second pulse position in each track. Moreover, by combining the content of the secret message with the pulse position in each track, the secret is inserted into the encoding process. To be specific, in the $k$-th track ($k = 0,1, \dots ,4$), $i_k$ and $i_{k+5}$ indicates the first and second pulse position. $(m)_{2k,2k+1}$ is the two embedded bits. For the two positions of the $i_{k+5}$, it satisfies the following equation [20]:

$$i_{k+5} = \begin{cases} ^{-1}\left(Z\left(\left\lfloor \frac{i_k}{5} \right\rfloor\right) \oplus (m)_{2k,2k+1}\right).5 + k \\ ^{-1}\left(Z\left(\left\lfloor \frac{i_k}{5} \right\rfloor\right) \oplus (m)_{2k,2k+1} + 4\right).5 + k \end{cases} , \tag{1}$$

where $Z$ indicates the encoding operations and $Z^{-1}$ is the decoding operations, which is based on table lookup, "$\oplus$" indicates the XOR operation, and $\lfloor x \rfloor$ indicates floor operation. Therefore, the secret message $(m)_{2k,2k+1}$ can be extracted as follows:

$$(m)_{2k,2k+1} = \left(Z\left(\left\lfloor \frac{i_k}{5} \right\rfloor\right) \oplus Z\left(\left\lfloor \frac{i_{k+5}}{5} \right\rfloor\right)\right) \bmod 4. \tag{2}$$

Based on Geiser's steganography method, adaptive information hiding is achieved by Miao et al. [21]. It brings in an embedding factor to controls the process of embedding, which is slightly different from Geiser's steganography method. To be specific, in order to embed the secret message $m_k$ into $k$-th track, the pulse position in $k$-th track should satisfy the following equation:

$$m_k = \left( \sum_{j=0}^{H_{k-1}} Z\left( \left\lfloor \frac{p_{k,j}}{N} \right\rfloor \right) \right) \oplus \eta,$$ (3)

where $H_k$ indicates the total number of non-zero pulses in the $k$-th track, $Z$ indicates the encoding operation, $N$ is the total number of the tracks for the specific speech encoding mode, and $\eta$ indicates the embedding factor. Therefore, the secret message can be extracted from Equation (3). For more detail about the above adaptive information hiding, please refer to [21].

### 2.2. Review of the State-Of-The-Art for AMR-Based Steganalysis

Steganalysis based on statistical characteristics of pulse pairs (SCPP) [33] for AMR speech includes three parts. In this section, we mainly pay our attention to the short-term features (STFS) and track-to-track features (TTFS), due to its connection to the Markov transition matrix.

To better illustrate the above features, we first introduce the pulse pair and the Markov transition matrix. There are $N$ subframes, with each subframe containing $T$ tracks. Specifically, a pulse pair $(p_{j,i}, p_{j,i+T})$ ($0 \leq i \leq T - 1$, $i$ indicates the $i$-th track) composed of two pulse positions can be extracted in terms of the $j$-th ($0 \leq j \leq N - 1$) subframe. In each track, for every pulse, there are $\tau$ candidate positions. Because of interchange in each track of AMR speech codec, $(p_{j,i}, p_{j,i+T})$ is equal to $(p_{j,i+T}, p_{j,i})$. Therefore, for each track, we can get the number of pulse pairs $\psi$:

$$\psi = \tau^2 - \frac{\tau(\tau - 1)}{2} = \frac{\tau^2 + \tau}{2}.$$ (4)

Grounded in the short-term invariance of speech signals, in the same track, the pulse pair of the current subframe is connected to the pulse pair of the prior subframe strongly. Precisely, in terms of the $i$-th track, each pulse pair can be represented as $g_{i,j}$ and the sequence of pulse-position pairs $G_i = \{g_{i,0}, g_{i,1}, \ldots, g_{i,N-1}\}$ is a Markov chain. Therefore, the first-order Markov chain of $G_i$ satisfies

$$P(g_{i,j}|g_{i,0}, g_{i,1}, \ldots, g_{i,j-1}) = P(g_{i,j}|g_{i,j-1}).$$ (5)

For all subframes in the $i$-th track, the probability $P((\gamma_1, \kappa_1)|(\gamma_2, \kappa_2))$ that the pulse pair $(\gamma_1, \kappa_1)$ occurs after the pulse pair $(\gamma_2, \kappa_2)$ is

$$P((\gamma_1, \kappa_1)|(\gamma_2, \kappa_2)) = P(g_{i,j} = (\gamma_1, \kappa_1)|g_{i,j-1} = (\gamma_2, \kappa_2)) = \frac{P(g_{i,j} = (\gamma_1, \kappa_1), g_{i,j-1} = (\gamma_2, \kappa_2))}{P(g_{i,j-1} = (\gamma_2, \kappa_2))}.$$ (6)

Naturally, the Markov transition matrix (denoted by $M_i$) is determined as follows:

$$M_i = \begin{bmatrix} P(\eta_{i,0}|\eta_{i,0}) & P(\eta_{i,0}|\eta_{i,1}) & \cdots & P(\eta_{i,0}|\eta_{i,\psi-1}) \\ P(\eta_{i,1}|\eta_{i,0}) & \ddots & & \\ \vdots & & \ddots & \\ P(\eta_{i,\psi-1}|\eta_{i,0}) & \cdots & \cdots & P(\eta_{i,\psi-1}|\eta_{i,\psi-1}) \end{bmatrix},$$ (7)

where $\eta_{i,t} = \left(\mu_{i,x}, \mu_{i,y}\right)$ indicates the *t*-th ($0 \leq t \leq \psi - 1$) pulse-position pair in terms of the *i*-th track. Then, for each pulse, assuming it exists $\tau$ candidate positions: $x$, $y$, and $t$ satisfy the equation:

$$t = \begin{cases} y, & x = 0, 0 \leq y \leq \tau - 1 \\ \sum\limits_{i=0}^{x-1} (\tau - i) + y - x & 1 \leq x \leq \tau - 1, 0 \leq y \leq \tau - 1 \end{cases}. \tag{8}$$

As mentioned above, in each track, the number of pulse-position pairs is $\psi$. Therefore, the dimension of each Markov transition matrix is $\psi \times \psi$. When taking all tracks into account, the dimension of the feature is high. To tackle this problem, the average Markov transition matrix is applied as the steganalysis features instead. Naturally, we can get the average Markov transition matrix (denoted by $M$) by calculating the following equation:

$$M = \frac{\sum\limits_{i=0}^{T-1} M_i}{T}. \tag{9}$$

Thus, the dimension of STFS is $\psi \times \psi$.

Moreover, with regard to pulse pairs in different tracks, joint probability matrices are applied to represent the TTFS feature. Concretely, for the pulse pair in different tracks, the joint probability matrix (JPM) $J_{i,m}$ is

$$J_{i,m} = \begin{bmatrix} P(\lambda_{i,0}, \lambda_{m,0}) & P(\lambda_{i,0}, \lambda_{m,1}) & \cdots & P(\lambda_{i,\psi-1}, \lambda_{m,\psi-1}) \\ P(\lambda_{i,1}, \lambda_{m,0}) & \ddots & & \\ \vdots & & \ddots & \\ P(\lambda_{i,\psi-1}, \lambda_{m,0}) & \cdots & \cdots & P(\lambda_{i,\psi-1}, \lambda_{m,\psi-1}) \end{bmatrix}, \tag{10}$$

where $\lambda_{i,h}\left(\lambda_{m,h}\right)$ indicates the *h*-th ($0 \leq h \leq \psi - 1$) pulse pair for the *i*-th (*m*-th) track. To be specific, for the joint probability of the pulse pair in the different tracks, it is calculated by the following equation:

$$P((\alpha_i, \beta_i), (\alpha_m, \beta_m)) = \frac{\sum\limits_{k=0}^{N-1} (\sigma(u_{k,i} = (\alpha_i, \beta_i)) \& (\sigma(u_{k,m} = (\alpha_m, \beta_m))))}{N}, \tag{11}$$

where $u_{k,i}\left(u_{k,m}\right)$ is the pulse pair in *i*-th (*m*-th) track of *k*-th subframe ($0 \leq k \leq N - 1$). Therefore, similar to STFS, the average JPM is adopted as TTFS sets to reduce the dimension of the feature and the training time for the classification model. The average JPM (marked by $J$) can be calculated by the following equation:

$$J = \frac{2 \sum\limits_{i=0}^{T-1} \sum\limits_{m=i+1}^{T-1} J_{i,m}}{T(T-1)}. \tag{12}$$

Thus, the dimension of TTFS is $\psi \times \psi$.

Finally, the total dimension of all two features is $2 \times \psi \times \psi$. In this paper, for the AMR-based speech codec, we assume it is at 12.2 kbps mode. In each subframe, it exists five tracks (i.e., $T = 5$). Eight candidate positions can be selected by two pulses (i.e., $\tau = 8$) on the same track. Therefore, $\psi$ is equal to 36, which is determined by Equation (4). Then, the total dimension is 2592. However, the dimension of these features is so high that some problems can be caused thereafter, such as the long training time of classification and the classification model being overfitting. To address this problem, Adaboost [34] is applied to feature selection in the feature set. Thus, the 498 features are acquired for AMR-based steganalysis.

From the above process, we can learn that the above steganalysis feature neglects the global property of the Markov chain. Thus, we can get the convergence feature representing the global



property of the Markov chain in the two aspects. First, for the pulse pair in the same track, we can extract the Markov chain from AMR speech. Second, for the pulse pair on a different track, we also can extract the Markov chain from AMR speech. Motivated by this, the convergence feature is proposed in our work. The convergence feature plus the SCPP feature can better characterize the AMR speech.

### 2.3. XGBoost Model

Compared to SVM (support vector machine), XGBoost (eXtreme Gradient Boosting) [41–46] has some advantages as follows. First, XGBoost can avoid overfitting, when the dimension of the feature is high. Second, XGBoost can mine the potential information from the hybrid features to provide better performance for classification. Last but not least, XGBoost is not sensitive to noise and thus has a better generalization ability. To have a better understanding of XGBoost, we first introduce the boosting algorithm; then we discuss decision trees; finally, we give a brief introduction to XGBoost.

#### 2.3.1. Boosting

The boosting algorithm is grounded in the weakly learnable theory, in which bias and variance can be reduced. Theoretically, weak learning algorithms can be combined into a strong one by a boosting algorithm. Its main idea is to enhance the learning algorithm for training samples that are easily misclassified so that the integrated classifiers subsequently focus on these samples. For example, Adaboost is the typical application of the boosting algorithm. The method of how to construct the integrated classifiers on the misclassified training samples and the hypothesis of integrated classifiers can be utilized to distinguish the difference between various boosting algorithms.

#### 2.3.2. Decision Trees

The decision tree is a flowchart-like structure. In terms of each internal node and each branch in a tree decision, it respectively corresponds to a test on an attribute and an output for the sample. Besides, each leaf node represents a category. The node at the top of the decision tree is the root node and contains a collection of all the data. Each internal node is a decision condition, and containing a dataset that satisfies all conditions from the root node to the current node. Thus, the dataset corresponding to the internal node is divided into two or more child nodes, and the number of branches is determined by the characteristics of the features on the internal node. To better elaborate on the decision tree, we take the C4.5 algorithm as an example. In C4.5 decision trees, information gain is adapted to select the splitting feature as a criterion. Each decision tree is constructed recursively.

#### 2.3.3. XGBoost

XGBoost is proposed by Tianqi Chen [43] and is an extensible decision tree-based boosting method. Its main idea is to continuously add decision trees and continuously perform feature splitting to grow a decision tree. Each added decision tree is equivalent to a new function to fit the residuals of the last prediction. The score of each sample relies on the characteristics of each sample. Each sample falls to a corresponding leaf node in each tree. Each leaf node corresponds to a score, and the predicted value of each sample can be obtained by the boosting method. To further elaborate XGBoost, it can be from the following aspects.

First, it is based on gradient boosting decision tree and its objective loss function is determined as follows,

$$\ell(\phi) = \sum_{i=1}^{n} loss(y_i, \hat{y}_i) + \sum_{k=1}^{K} \Omega(f_k), \tag{13}$$

where $l(\phi)$ indicates the training loss, $\Omega(f)$ controls the complexity of the decision tree to prevent the classification model from overfitting, and $K$ indicates the number of trees. By minimizing this objective function, this model can achieve an optimal state. To tackle this problem, an additive training method is applied to slow training loss, and Taylor expansion is adapted to optimize the prediction at the

additive $t^{th}$ training round. The optimal complexity of each decision tree could be solved by the greedy algorithm.

Moreover, to improve the performance of XGBoost [41–46], several supernumerary technologies are applied in XGBoost. First, compared to the traditional decision tree-based integrated learning method [44], it adds the regularity items to the loss function and the regularity items include the weight of each leaf node and the single tree complexity. By assigning a weight to each leaf node and every single tree, XGBoost can exploit the correlation of combined features. The single tree complexity is used to avoid overfitting (overfitting means making assumptions too strict to get consistent assumptions). In addition to the regularized technology in the objective loss function, Column subsampling and shrinkage are also utilized to avoid overfitting. Shrinkage updates newly added weights for each step of tree boosting, by bringing in a factor $\alpha$. By reducing the weight of every single tree, shrinkage remains an improvement space for the upcoming trees. This idea is similar to a learning rate in stochastic optimization. Column subsampling is equivalent to feature selection, which means that the number of the variable used to establish a decision tree is reduced. The fewer the number of variables in each decision tree is, the simpler each decision tree is. In other words, XGBoost can avoid overfitting. Moreover, column subsampling can speed up computation time. Finally, XGBoost supports the parallel calculation of the gain of each feature and then selects the optimal feature for splitting.

## 3. Proposed Scheme

For this work, the motivation behind our presented scheme is as follows. First, the convergence feature can reflect the global and stable features of the AMR speech while the statistical characteristics of pulse pairs (SCPP) [33] can only represent the local and instantaneous features. On account of this, the convergence feature combined with the SCPP feature [33] can better represent the characteristics of AMR speech. Second, compared to the SVM classifier adapted by the previous study [31–33], XGBoost enjoys an advantage in preventing the classification model from being overfitting and generalization ability. Finally, XGBoost can mine the potential information from the hybrid feature, which includes the SCPP feature and the convergence feature.

### 3.1. Convergence Features Based on Markov Chain

The Markov chain (MC) will converge to a stable probability distribution by multiplying the Markov transition matrix several times, which can reflect the global characterization of the Markov chain, according to [45–47]. This property has been applied to many types of research, such as capturing the dynamics embodied in the data [45], calculating the regional energy efficiency [46], and optimization problems [47].

The Markov chain is a Markov process with discrete-time and state parameters. In a random process $U = \{u_1, u_2, \ldots, u_N\}$, the system state set is recorded as $S = \{s_1, s_2, \ldots, s_n\}$. For the first-order of $U$, it satisfies

$$
\begin{aligned}
P \quad & \{u_{N+1} = s_{n+1} | u_N = s_n, u_{N-1} = s_{n-1}, \ldots, u_1 = s_1\} \\
& = P\{u_{N+1} = u_{n+1} | u_N = s_n\}.
\end{aligned}
\tag{14}
$$

State transition probability refers to the probability of one state going to another state. The probability $P(s_j|s_i)$ that the state $s_j$ occurs after the state $s_i$ is

$$
P(s_j|s_i) = \frac{P(u_{m+1} = s_j, u_m = s_i)}{P(u_m = s_i)}.
\tag{15}
$$

Moreover, the Markov transition matrix of $U$ (denoted as $D$) is determined as follows:

$$D = \begin{bmatrix} P(s_1|s_1) & P(s_2|s_1) & \cdots & P(s_n|s_1) \\ P(s_1|s_2) & \ddots & & \\ \vdots & & \ddots & \\ P(s_1|s_n) & \cdots & \cdots & P(s_n|s_n) \end{bmatrix}. \tag{16}$$

It satisfies the following properties,

$$\begin{aligned} 0 &< D_{ij} < 1 \\ \sum_{i=1}^{n} D_{ij} &= 1. \end{aligned} \tag{17}$$

Assume that there exists a probability distribution $\pi = \{\pi_i, i \in S\}$, it satisfies the following condition,

$$\pi(j) = \sum_{i \in S} \pi(i) D_{ij}, \tag{18}$$

where $\pi$ is called the smooth distribution of $U$. Moreover, if $\pi(j)$ is not related to $i$, it can be defined as

$$\lim_{n \to \infty} (D_{ij})^n = \pi(j). \tag{19}$$

According to Equation (18), we can get

$$\lim_{n \to \infty} D^n = \begin{bmatrix} \pi(1) & \pi(2) & \ldots & \pi(j) & \ldots \\ \pi(1) & \pi(2) & \ldots & \pi(j) & \ldots \\ \ldots & \ldots & \ldots & \ldots & \ldots \\ \pi(1) & \pi(2) & \ldots & \pi(j) & \ldots \\ \ldots & \ldots & \ldots & \ldots & \ldots \end{bmatrix}. \tag{20}$$

Therefore, we have the following definition

$$D_c = \lim_{n \to \infty} D^n. \tag{21}$$

By incorporating Equations (19) and (20), $D_c$ will reach a smooth state, which indicates $D_c$ is convergent. According to Equation (20), $D_c$ as a convergence matrix, is also called the convergence feature. Moreover, the convergence matrix is symmetry and each column has the same value. Moreover, we can select one row from the convergence matrix as the convergence feature.

## 3.2. Analysis of the Combination of Convergence Feature and Statistical Characteristics of Pulse Pairs

Information fusion is a technique, which combines the information from various sources with diverse conceptual, vision, and time representations as an integral whole. Its goal is aimed at achieving better performance, compared to the disparate source [48]. This technology has been applied to logic-based fusion [49] and remote sensing [50]. For the type of information fusion, it can be roughly classified into three types, including feature level, decision level, and score level. In our work, feature fusion is applied to represent AMR speech. For the feature fusion, different features are combined as an integral whole. The combined feature is used to train the classifier model.

In our scheme, the combined feature includes two parts. The first part is the SCPP feature [33]. It reflects the local feature of the AMR speech. The second part is the convergence feature proposed in our work. It reflects the global feature of the AMR speech. the Markov transition matrix can be extracted from the AMR-based speech streams. The convergence feature is a useful complement to the SCPP feature. Thus, grounded in the above analysis, we combine the SCPP and convergence features

as final AMR speech steganalysis features for classification and the combined feature can improve the performance of AMR speech steganalysis.

### 3.3. XGBoost-Based Steganalysis Scheme

As for the proposed scheme, it adopts statistical features plus classifier. XGBoost is adapted as the classifier, which is the one contribution in our scheme, compared to the existing study [31–33] using SVM as the classifiers. The extracted steganalysis feature is based on AMR speech codec at 12.2 kbps. The steganalysis feature includes two parts. The first part is the SCPP feature, which includes the short-term feature mentioned in Section 2.2 and the track-to-track feature mentioned in Section 2.2. And the dimension of SCPP feature is 498. The second part is the convergence feature proposed in our scheme, which is another contribution to our method. We can get the convergence feature from the average Markov transition matrix used to characterize short-term feature and its dimension is 36. Similarly, we also get another convergence feature from the average Markov transition matrix employed to characterize the track-to-track feature and its dimension is 36. Therefore, the total dimension of the convergence feature is 72 and the dimension of the combined feature is 570. Finally, the steganalysis features vary dynamically with different AMR speech codecs.

The training phase is as follows. First, the SCPP feature [33] mentioned in Section 2.2 is extracted from the AMR speech. Then, the convergence feature is extracted from the AMR speech (denoted as CG), according to the above method mentioned in Section 2.1. Moreover, two features are merged into an integrated feature (denoted as SCPP-CG). Finally, the integrated feature is sent to XGBoost to train the classification model. The framework of our presented scheme is shown in Figure 1, which includes the training phase and the testing phase.

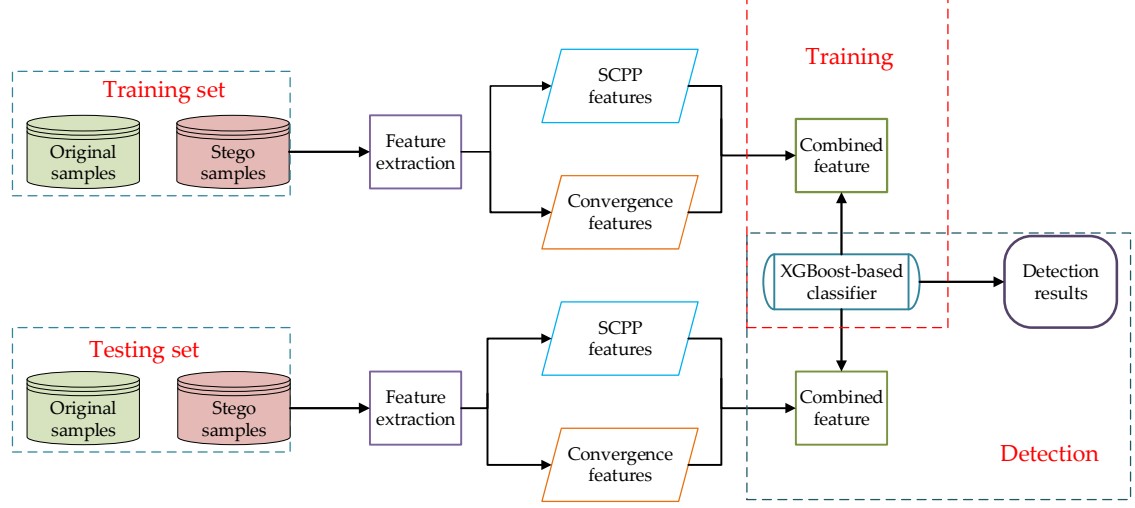

**Figure 1.** The proposed extreme gradient boosting (XGBoost)-based steganalysis scheme with a combined feature.

## 4. Experimental Result and Analysis

This section is organized as follows. First, we introduce our experimental circumstances, dataset, and metrics that are applied to evaluate our presented scheme. Then, to evaluate our method with the state-of-the-art, the accuracies, false-positive rate (FPR), and false-negative rate (FNR) are applied to evaluate the performance. Finally, we have an analysis of the experimental results.

### 4.1. Experimental Setup, Data, and Metrics

To compare our work to the state-of-the-art, libSVM was applied to achieve the SVM classification model in the same condition. Specifically, the type of SVM is C-style. For all parameters of the SVM

classification model based on the RBF kernel, they adopted the default settings. The scikit-learn python libraries were implemented to achieve XGBoost-based classifiers. In our experiments, we only adjusted the value of our parameters and other parameters were set with default parameters. To make the XGBoost classification model more stable and have a better generalization ability, we set the learning rate as 0.01. For the number of booted trees, we set it to 250. The subsample was set to 0.7, which can prevent the classification model from overfitting. For the maximum-depth of each tree in this experiment, it was set to 6, which can prevent the model from being complex. With regard to the other parameter, the default values were adapted. For the value of all parameters, we used the Bayesian optimization algorithm to obtain its optimal values.

As is known to all, in our research field, there is no public steganography/steganalysis dataset. To assess the presented scheme objectively, we have to establish our dataset with two subsets. Specifically, the cover speech dataset represents positive samples and the steganographic speech dataset represents negative samples. To deal with this problem, we collected 8000 ten-second speech samples from audio materials from the Internet. Those speech samples were made up of different speakers. To evaluate our algorithm objectively, the 12.2 kbps mode in AMR codec was selected as the cover and the cover speech dataset was composed of those speech samples. To verify our experiment objectively, the length of the AMR-based speech sample was ten-seconds, according to the previous study [33].

To construct a steganographic speech dataset, randomly generated bitstreams were embedded into each sample in the cover speech dataset by using the steganography methods [20,21]. Then, we defined the embedding rate, which indicates the proportion of the number of embedding bits to the whole capacity of the cover sample. Moreover, for each sample in the cover speech dataset, we conducted four steganography methods [20,21] to it with the embedding rate ranging from 10% to 100% at a step of 10%. Finally, the steganographic speech dataset was composed of the generated speech samples.

In this work, we present two methods to verify the rationality of our ideas and compare them with the state-of-the-art method [33]. The state-of-the-art method (denoted as SCPP+SVM) adopts SCPP as the steganalysis feature and SVM as the classifier. Moreover, the first method adopts SCPP as the steganalysis feature and XGBoost as the classifier (denoted as SCPP + XGBoost). It was applied to verify the idea that XGBoost can avoid overfitting when the dimension of the steganalysis feature is high, compared to the state-of-the-art method. The second method is our final proposed method (denoted as the proposed method), which adopts both the SCPP feature and the convergence feature as the steganalysis feature and XGBoost as the classifier. The second method can verify that the convergence feature is a useful complement to the SCPP feature, compared to the first method.

Furthermore, we randomly chose 4000 samples from the steganographic segments dataset and 4000 samples from the cover speech dataset. Those 8000 samples were composed of the training set. Then, the remainder 4000 samples in cover samples and the remainder 4000 samples in the steganographic segment samples make up the testing set.

To evaluate our proposed method with an objective standard, accuracy (ACC), FPR, and false FNR were adopted as evaluation standards, which are adopted by previous studies [32,33] and many machine learning algorithms. The accuracy is the proportion of the number of samples that are classified correctly to the amount of all samples. FPR is the proportion of false-positive samples to negative samples. FNR is the proportion of false-negative samples to all positive samples. Moreover, the receiver operating characteristic curve (ROC) is applied to evaluate the stability of our presented scheme.

## 4.2. Comparison of the Presented Scheme and Existing Ones

In this experiment, to evaluate our present scheme, the length of the AMR-based speech sample is set to ten-seconds according to the preceding study [33]. Then, we further made comparisons to the previous methods (i.e., SCPP [33] with SVM and Fast-SPP [32]) by conducting a series of experiments

with embedding rates ranging from 10% to 100% at the step of 10%. In this research, we do not consider Fast-SPP, since SCPP was proven to be better.

The experimental results are shown in Figures 2–5, which respectively correspond to four different steganography methods at different embedding rates. From these figures, there come several conclusions as follow. First, with the embedding rate increasing, the accuracies of all three steganalysis schemes are increasing. Second, the proposed method outperforms SCPP based on SVM [33] and SCPP based on XGBoost, not only on the accuracy but also on the FPR and FNR. Third, the SCPP based on the XGBoost scheme presents some merits of verification advantages compared to the SCPP based on the SVM scheme. Lastly, when the embedding rate is larger than 40%, the accuracies of the three steganalysis methods are almost the same.

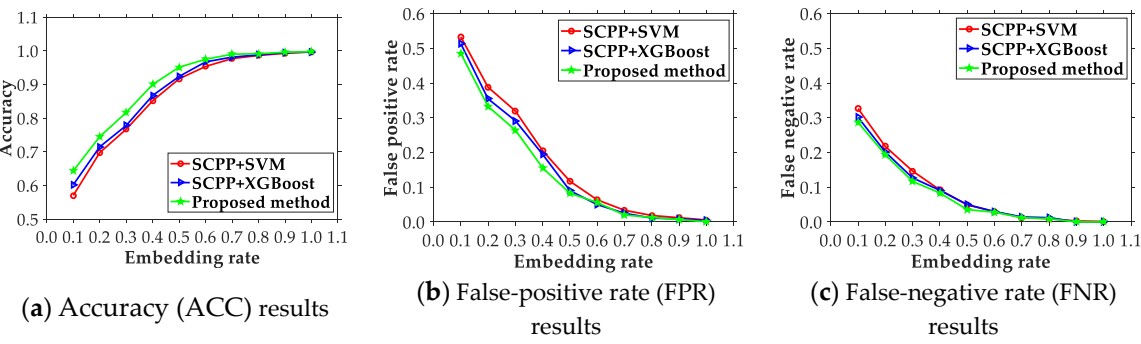

**Figure 2.** Experimental Results for Geiser's method.

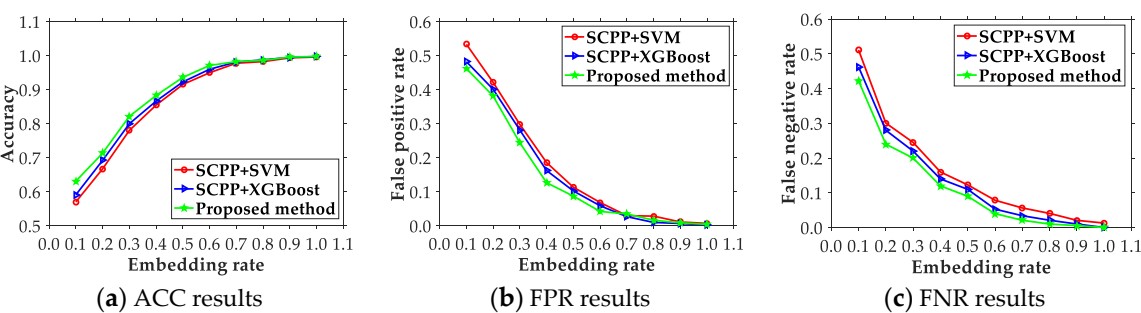

**Figure 3.** Experimental Results for Miao's method ($\eta = 1$).

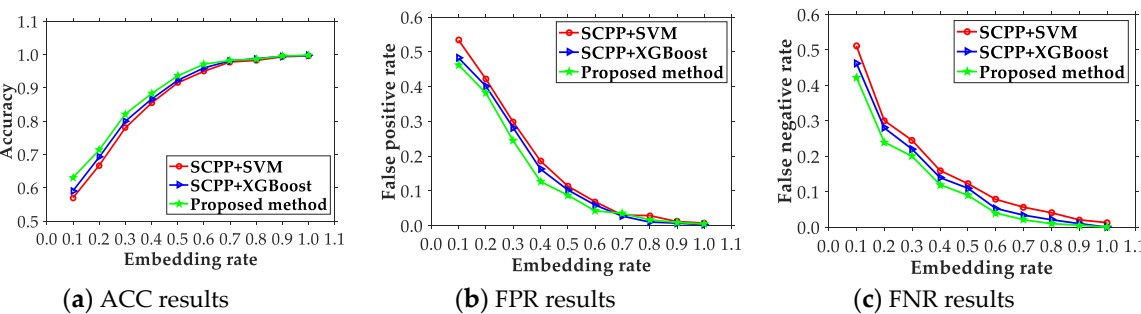

**Figure 4.** Experimental Results for Miao's method ($\eta = 2$).

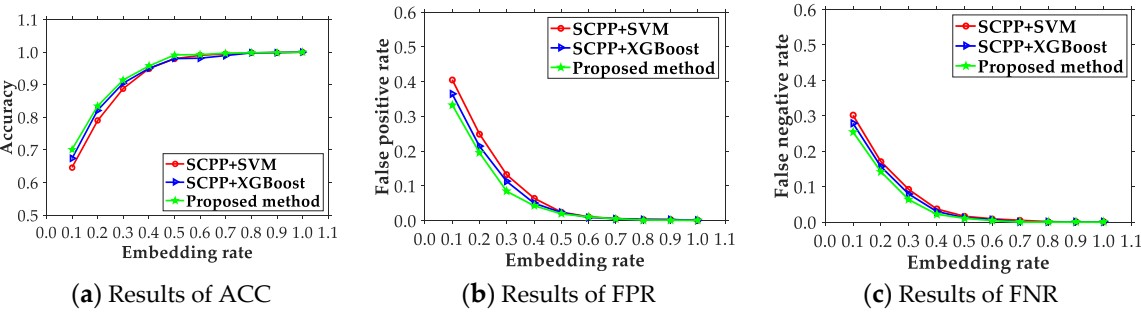

(**a**) Results of ACC　　　　(**b**) Results of FPR　　　　(**c**) Results of FNR

**Figure 5.** Experimental results for detecting Miao's method ($\eta = 4$).

For the above results, we can explain it from several aspects. First, the proposed method based on the combined feature can better characterize the feature of AMR-based speech. The combined feature characterizes the dynamic feature and stable feature of AMR-based speech. Second, XGBoost is applied to classification, which boasts advantages in preventing the classification model from overfitting and having good generalization ability. Moreover, XGBoost can mine the potential information from the combined feature, according to its property, which can provide some more useful information for the classification, compared to SVM. Finally, for the embedding rate smaller than 40%, the embedding capacity of ten-second speech samples is hard to be classified while our presented scheme outperforms the SCPP based on the SVM scheme [33]. When the embedding rate is greater than 40%, the steganography samples can be easily distinguished, due to larger embedding capacities.

To make our proposed scheme more convincing, the receiver operating characteristic (ROC) curve is applied to verify the performance of our method, by comparing it with the state-of-the-art methods. Without a loss of generality, we select several embedding rates of 30%, 60%, and 100% for all four steganographic methods, according to the preceding study [33]. The results are shown in Figures 6–9. We can conclude that the performance of our presented scheme is better than the existing steganalysis scheme for the existing steganographic methods.

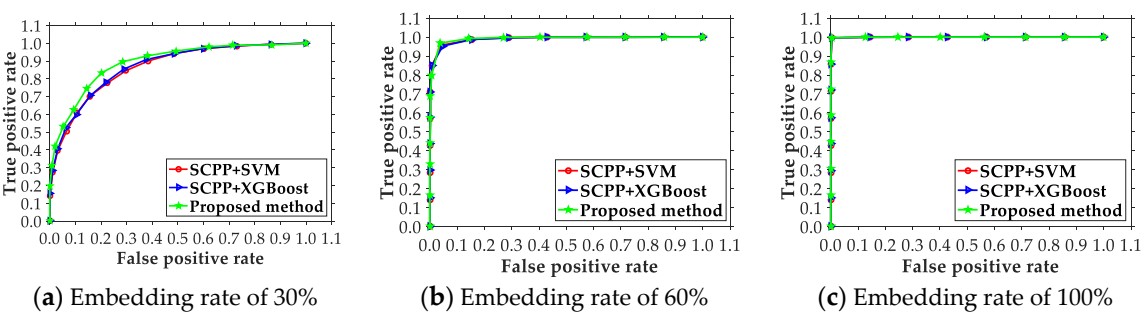

(**a**) Embedding rate of 30%　　　　(**b**) Embedding rate of 60%　　　　(**c**) Embedding rate of 100%

**Figure 6.** The ROC curves for detecting Geiser's method.

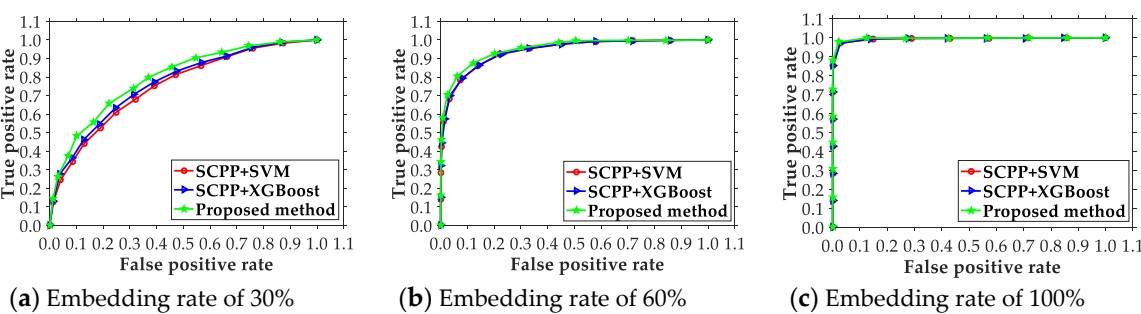

(**a**) Embedding rate of 30%　　　　(**b**) Embedding rate of 60%　　　　(**c**) Embedding rate of 100%

**Figure 7.** The ROC curves for detecting Miao's method ($\eta = 1$).

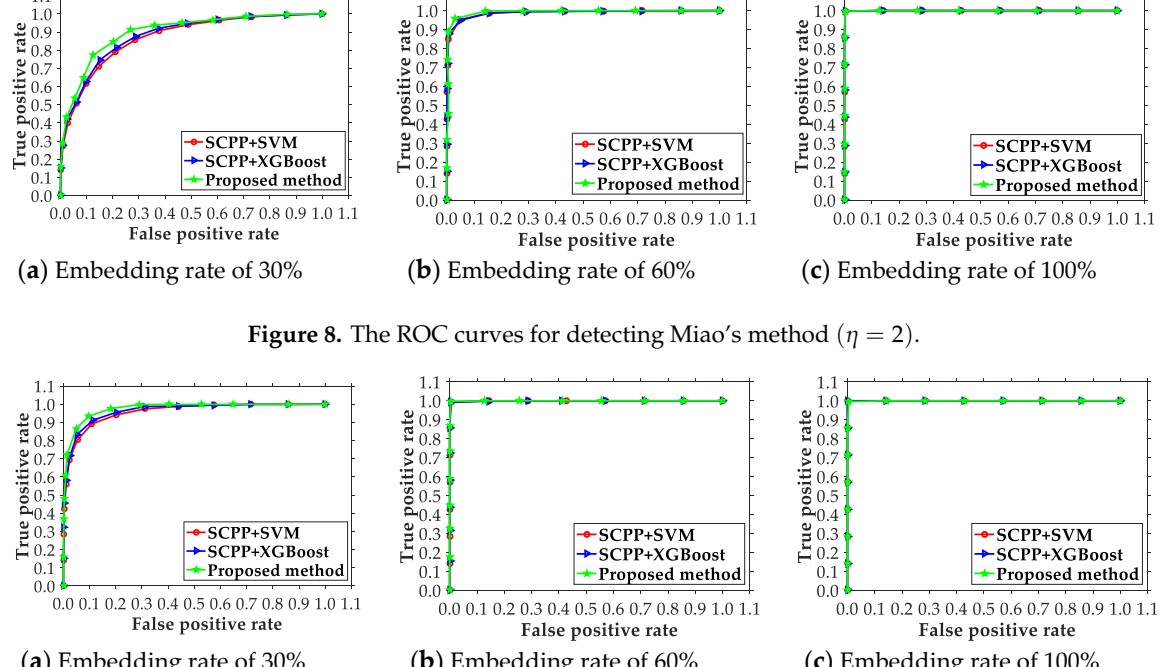

**Figure 8.** The ROC curves for detecting Miao's method ($\eta = 2$).

**Figure 9.** The ROC curves for detecting Miao's method ($\eta = 4$).

## 5. Conclusions

In our modern world, voices over the network are becoming more increasingly popular and widespread. As a result, some relevant steganographic techniques have been successfully developed. However, those steganography techniques may be a tool for criminals to commit crimes. Under such circumstances, it is worth studying the countermeasure, steganalysis of AMR-based speech. According to previous research on steganalysis, the feature extracted from AMR-based speech is almost from the Markov transition matrix, which cannot characterize all the features of AMR-based speech. For that matter, we are motivated by the nature of the Markov transition matrix and propose the convergence feature of AMR-based speech streams. Then we combine the convergence feature with the SCPP feature as the final feature for the detection with XGBoost applied to the final classification. The experimental results indicate that our proposed scheme can provide better performance for detecting the existing steganographic methods in AMR-based speech streams. In the future, we will focus on how to combine the different features more effectively and select a better classification model for the combined feature. Moreover, we will improve the proposed method by data preparation techniques, such as clustering [51–53], normalization, and data cleaning.

**Author Contributions:** All authors contributed to the preparation of this paper. H.T. and C.S. proposed the method, performed the literature review, guided the experiments, analyzed the results, and wrote the manuscript; C.S. designed and conducted the experiments; C.-C.C. supervised the research work and provided suggestions for improvement; Y.C. (Yewang Chen), Y.D., Y.-H.C., C.C.C. and Y.C. (Yiqiao Cai) analyzed the experimental results and provided revising suggestions. All authors have read and agreed to the published version of the manuscript.

**Funding:** This research was funded by National Natural Science Foundation of China under Grant Nos. 61972168 and U1536115, Natural Science Foundation of Fujian Province of China under Grant No. 2018J01093, Opening Program of State Key Laboratory of Information Security of China under grant No. 2019-ZD-09, and Subsidized Project for Postgraduates' Innovative Fund in Scientific Research of Huaqiao University No. 17014083010.

**Acknowledgments:** The authors sincerely thank the anonymous reviewers for their constructive comments that greatly improved the manuscript.

**Conflicts of Interest:** The authors declare no conflict of interest regarding the publication of this paper

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
