# Peer review of "Steganalysis of Adaptive Multi-Rate Speech Based on Extreme Gradient Boosting"

_electronics, doi:10.3390/electronics9030522_

Round 1

Reviewer 1 Report

The paper is about the combination of the convergence feature of AMR-based speech streams with the SCPP as the final feature for the detection with XGBoost classification applied to the final classification. The work is well organized and the results are significant. 

However, the work needs significant improvement, since in some points it is difficult to read. Specifically, the introduction needs to be rewritten. It is important that there exist no grammatical errors (e.g., propose instead of purpose, background and preliminary instead of "...preliminary knowledge" etc.) and that the text is comprehensive and that it follows a flow. The paragraphs seem to be unconnected and the reader should understand the logic leap from one paragraph to the other. I strongly suggest rewritting the Introduction Section.

At first appearance of a term, a description or anything else related should be provided. For example, in line 64, there is a reference to η, which should be followed also by its name (eta), and the same stands for ψ (psi). Although these characters should be known by everyone, the audience is international and we should make no assumptions for their knowledge level. What does SCPP stands for (it isn't  an abbreviation derived from some words)?

Equation 5 doesn't fit to one line? The same stands for eq. 10.

Lines 193-194, sentence doesn't make sense "Theoretically, for a weak learning algorithm, it is capable of a strong learning algorithm by the Boosting algorithm"

Please avoid sentences-paragraphs like in lines 208-209. It should be part of the previous paragraph.

Author Response

      Responses to Reviewers' Comments on Electronics-676462

Congcong Sun, Hui Tian, Chin-Chen Chang, Yewang Chen, Yiqiao Cai,

Yongqian Du, Yonghong Chen, Chih-Cheng Chen

Dear Editor,

We thank you very much for handling our manuscript with useful suggestions. We also thank the reviewers for the valuable time and efforts spent in reviewing our paper and helping us improve it. After careful review and consideration of all comments, we have addressed all concerns that were raised and made corresponding modifications in this revision. The following are detailed point-by-point responses to reviewers' comments.

Reviewer #1:

Comments (1.1):

The paper is about the combination of the convergence feature of AMR-based speech streams with the SCPP as the final feature for the detection with XGBoost classification applied to the final classification. The work is well organized and the results are significant. However, the work needs significant improvement, since at some points it is difficult to read. Specifically, the introduction needs to be rewritten. It is important that there exist no grammatical errors (e.g., propose instead of purpose, background and preliminary instead of "...preliminary knowledge" etc.) and that the text is comprehensive and that it follows a flow. The paragraphs seem to be unconnected and the reader should understand the logic leap from one paragraph to the other. I strongly suggest rewriting the Introduction Section.

Response to Reviewer Comment (1.1):

Thank you for your encouraging comments and constructive suggestions. According to your valuable suggestions, we have carefully revised the corresponding portions of this manuscript. The Introduction Section has been rewritten. Specifically, the first paragraph introduces steganography in brief, then gives a brief description to its applications and history (why many researchers have devoted to steganography), finally describes the reason why VoIP is considered as an ideal steganographic carrier, compared to other ones; the second paragraph introduces the wide application of AMR speech in VoIP (why the AMR speech is selected as the main steganographic carrier in VoIP); the third paragraph introduces the AMR speech-based steganography; the fourth paragraph elaborates the reason why we study the steganalysis of AMR speech; the fifth paragraph reviews the existing work related to our research; the sixth paragraph analyzes the drawbacks of previous work; the final paragraph elaborates the main idea of the presented scheme and our contributions.

In addition, we try our best to correct grammatical errors. Especially, “purpose” has been modified to “propose”; "...preliminary knowledge" has been modified to “background and preliminary”.

We hope that our effort could further improve the readability of our paper.

Comment (1.2):

At first appearance of a term, description or anything else related should be provided. For example, in line 64, there is a reference to η, which should be followed also by its name (eta), and the same stands for ψ (psi). Although these characters should be known by everyone, the audience is international and we should make no assumptions for their knowledge level. What does SCPP stand for (it isn't an abbreviation derived from some words)? Equation 5 doesn't fit one line? The same stands for eq. 10. Lines 193-194, the sentence doesn't make sense "Theoretically, for a weak learning algorithm, it is capable of a strong learning algorithm by the Boosting algorithm" Please avoid sentences-paragraphs like in lines 208-209. It should be part of the previous paragraph.

Response to Reviewer Comment (1.2):

Thank you for your valuable advice. We have made the correction, according to your comments. We have provided the names or meanings for all the involved variables in this paper. Particularly, the variable “η” is the embedding factor; the variable “ψ” is the number of pulse pairs. The corresponding revision can be found in lines 63~64 and line 158.  SCPP is the abbreviation derived from “statistical characteristics of pulse pairs”, which can be found in lines 271-272. We have made modifications for Eq (5) and (10) as per your instruction. Now they are written on the same line. We have modified the sentence as “Theoretically, weak learning algorithms can be combined into a strong one by a boosting algorithm”. The modifications can be found in lines 218-219. We have corrected them as per your instruction, which can be found in lines 233-235.

Reviewer 2 Report

This paper proposed a steganalysis of adaptive multi-rate (AMR) which is used to secure the speech streams. The CG (extract convergence feature from AMR speech) and SCPP (extract steganalysis features) are used to extracts and combines features where XGBoost classification is applied to facilitate the high-dimensional features and avoid overfitting. As a result, the proposed method compared with other two existing methods slightly outperforms on each evaluated metric.

It is quite difficult to find novelty in the paper as all the techniques applied in this research is all developed in prior research and it is quite difficult to find new insight in the field. Moreover, the writing and flow of the content makes the paper very difficult to read and understand. It is also quite difficult to find the objectives and contributions in the paper. Followings are few comments about the contents indicating the above :

How the overfitting is prevented by XGBoost algorithm in the experiment need to be fully justified. The paper presents the usage of XGBoost to prevent overfitting when the dimension of feature is high. But there is no details about how many features are in combined features and whether the dimension of feature (in different audio) is fixed or dynamic, etc… Need more information about the extracted feature to fully understand and justify the proposed approach. The paper also lacks detailed explanation on SCPP. The section on XGBoost-based steganalysus scheme is very difficult to understand. Geiser’s method and Miao’s method are not explained.

Line #40 ~ #49 : There is short explanation on steganography but it does not help readers understand the steganography. Line #116 : Full names for abbreviation AMR-NB should be given. And this applied to all abbreviation used in the paper. Line #207 : The C4.5 algorithm is taken as an example but C4.5 is very briefly described. Then random forest is introduced in detail. Hence, it is quite difficult to see which algorithm the authors want to elaborate? Equation 12, 13, 14 seems to be just duplications with Equation 4, 5, 6. Line #230 & #237 : Reference is not cited in consistent manner. Line #299 ~ #306 : The machine learning’s basic training and testing procedure are described where there is quite a duplication in the descriptions.

Author Response

      Responses to Reviewers' Comments on Electronics-676462

Congcong Sun, Hui Tian, Chin-Chen Chang, Yewang Chen, Yiqiao Cai,

Yongqian Du, Yonghong Chen, Chih-Cheng Chen

Dear Editor,

We thank you very much for handling our manuscript with useful suggestions. We also thank the reviewers for the valuable time and efforts spent in reviewing our paper and helping us improve it. After careful review and consideration of all comments, we have addressed all concerns that were raised and made corresponding modifications in this revision. The following are detailed point-by-point responses to reviewers' comments.

Reviewer #2:

Comment (2.1):

This paper proposed a steganalysis of adaptive multi-rate (AMR) which is used to secure the speech streams. The CG (extract convergence feature from AMR speech) and SCPP (extract steganalysis features) are used to extracts and combines features where XGBoost classification is applied to facilitate the high-dimensional features and avoid overfitting. As a result, the proposed method compared with other two existing methods slightly outperforms on each evaluated metric.

It is quite difficult to find novelty in the paper as all the techniques applied in this research is all developed in prior research and it is quite difficult to find new insight in the field. Moreover, the writing and flow of the content makes the paper very difficult to read and understand. It is also quite difficult to find the objectives and contributions in the paper.

Response to Reviewer Comment (2.1):

Thank you for your valuable advices. According to your suggestions, we have carefully revised the whole paper and tried our best to improve the readability of the paper. In addition, the purpose of our paper is to detect whether the secret information has been inserted into AMR speech streams. Our contributions can be summarized as follows. Firstly, we propose a convergence feature for detecting the FCB-based steganography, which describes global characteristics of AMR speech streams. Since the state-of-the-art feature SCPP mainly reflects local characteristics of AMR speech streams, the presented convergence feature is a useful complement to SCPP. Secondly, we propose a new steganalysis feature using both the convergence feature and SCPP. Finally, we present a new steganalysis model for AMR speech, which employs XGBoost rather than traditional SVMs employed in previous schemes as the classifier. It enjoys some advantages, such as mining the potential information from the hybrid features, avoiding overfitting (making assumptions too strict to get consistent ones) and having a strong generalization ability. The experimental results demonstrate that the presented method significantly outperforms the state-of-the-art one that adopts SVM as the classifier and SPCC as the feature. Therefore, we think that our work is significant and novel enough. The corresponding revision can be found in lines 103~113.

Comment (2.2):

Followings are few comments about the contents indicating the above :

How the overfitting is prevented by XGBoost algorithm in the experiment need to be fully justified. The paper presents the usage of XGBoost to prevent overfitting when the dimension of feature is high. But there is no details about how many features are in combined features and whether the dimension of feature (in different audio) is fixed or dynamic, etc… Need more information about the extracted feature to fully understand and justify the proposed approach. The paper also lacks detailed explanation on SCPP. The section on XGBoost-based steganalysus scheme is very difficult to understand. Geiser’s method and Miao’s method are not explained. Line #40 ~ #49 : There is short explanation on steganography but it does not help readers understand the steganography. Line #116 : Full names for abbreviation AMR-NB should be given. And this applied to all abbreviation used in the paper. Line #207 : The C4.5 algorithm is taken as an example but C4.5 is very briefly described. Then random forest is introduced in detail. Hence, it is quite difficult to see which algorithm the authors want to elaborate? Equation 12, 13, 14 seems to be just duplications with Equation 4, 5, 6. Line #230 & #237 : Reference is not cited in consistent manner. Line #299 ~ #306 : The machine learning’s basic training and testing procedure are described where there is quite a duplication in the descriptions.

Response to Reviewer Comment (2.2):

We have conducted experiments about the XGBoost algorithm to show how it can prevent overfitting. In this work, we present two methods to verify the rationality of our ideas and compare it with the state-of-the-art method. The state-of-the-art method (denoted as SCPP+SVM) adopts SCPP as the steganalysis feature and SVM as the classifier. Moreover, the first method adopts SCPP as the steganalysis feature and XGBoost as the classifier (denoted as SCPP + XGBoost). It is applied to verify the idea that XGBoost can avoid overfitting when the dimension of the steganalysis feature is high, compared to the state-of-the-art method. The second method is our final proposed method (denoted as proposed method), which adopts both the SCPP feature and the convergence feature as the steganalysis feature,and XGBoost as the classifier. The second method can verify that, the convergence feature is a useful complement to the SCPP feature, compared to the the first method. The corresponding revision can be found in lines 380~388. The steganalysis feature includes two parts. The first part is the SCPP feature, which includes the short-term feature mentioned in section 2.2 and the track-to-track feature mentioned in section 2.2. And its dimension is 498. The second part is the convergence feature proposed in our scheme, which is another contribution to our method. We can get the convergence feature from the average Markov Transition Matrix used to characterize short-term feature and its dimension is 36. Similarly, we also get another convergence feature from the average Markov Transition Matrix employed to characterize the track-to-track feature and its dimension is 36. Therefore, the total dimension of the convergence feature is 72 and the dimension of the combined feature is 570. Finally, The steganalysis features vary dynamically with different AMR speech codecs. The corresponding revision can be found in lines 331~340. The SCPP is the abbreviation derived from “statistical characteristics of pulse pairs”, of which the description can be found in lines 271~272. We have rewritten the section about the XGBoost-based steganalysis scheme. Particularly, it first gives detailed information about the classifier and steganalysis feature, then shows a detailed steganalysis scheme. Geiser’s method and Miao’s method are respectively the existing steganographic methods presented by Geiser and Miao,whose introductions can be found in Section 2.1. For line #40 ~ #49, it has been rewritten. Specifically, the first paragraph introduces steganography in brief, then gives a brief description to its applications and history (why many researchers have devoted to steganography), finally describes the reason why VoIP is considered as an ideal steganographic carrier, compared to other ones. We have revised theportion as per your instruction. Steganography is a security technique of embedding secret information into a certain carrier while the secret information can be extracted accurately. Thus, this technology can be applied to covert communication. AMR-NB is the abbreviation of Adaptive Multi-Rate – Narrowband. The corresponding revision can be found in lines 128-129. We have double-checked all abbreviations and provided their full names. We mainly want to elaborate on XGBoost. However, as the preliminary knowledge, we briefly introduce the C4.5 algorithm and the random forest. To further improve the readability, we re-organized our description.The corresponding revision can be found in lines 233~235 and lines 237~268. (14)-(16) reflect the more universal nature of the markov chain, not just for AMR speech steganalysis. This nature can be applied to other fields, which is also the value of this research. Therefore, we think that it is better to keep them in this section.   Wehave double-checked all the references and listed them in the required manner. We have deleted the reduplicated descriptionsfor the basic training and testing procedures of machine learning. The corresponding revision can be found in the section on the XGBoost-based steganalysis scheme.

Reviewer 3 Report

The steganographic techniques have been successfully developed due to the widespread of network voices but, on the other hand, it may be an effective tool to commit cybercrimes. Steganalysis of adaptive multi-rate speech is a hot topic for controlling cybercrimes grounded in steganography in related speech streams and a large number of researchers have been regarding AMR as ideal carriers for information embedding. The study proposes an extreme gradient boosting (XGBoost) based steganalysis scheme for adaptive multi-rate speech streams and results indicate that the proposed scheme can provide better performance for detecting the existing steganographic methods in AMR-based speech streams. It does present a very recent and important bibliography in the field and the methodology is clearly explained.

Author Response

      Responses to Reviewers' Comments on Electronics-676462

Congcong Sun, Hui Tian, Chin-Chen Chang, Yewang Chen, Yiqiao Cai,

Yongqian Du, Yonghong Chen, Chih-Cheng Chen

Dear Editor,

We thank you very much for handling our manuscript with useful suggestions. We also thank the reviewers for the valuable time and efforts spent in reviewing our paper and helping us improve it. After careful review and consideration of all comments, we have addressed all concerns that were raised and made corresponding modifications in this revision. The following are detailed point-by-point responses to reviewers' comments.

Reviewer #3:

Comment (3.1):

The steganographic techniques have been successfully developed due to the widespread of network voices but, on the other hand, it may be an effective tool to commit cybercrimes. Steganalysis of adaptive multi-rate speech is a hot topic for controlling cybercrimes grounded in steganography in related speech streams and a large number of researchers have been regarding AMR as ideal carriers for information embedding. The study proposes an extreme gradient boosting (XGBoost) based steganalysis scheme for adaptive multi-rate speech streams and results indicate that the proposed scheme can provide better performance for detecting the existing steganographic methods in AMR-based speech streams. It does present a very recent and important bibliography in the field and the methodology is clearly explained.

Response to Reviewer Comment (3.1):

Thank you very much for your positive comments.

Round 2

Reviewer 2 Report

All the comments given in the previous review was taken into account and adequate revision has been performed. But there are still minor issues which needs to be addressed as indicated below. Also careful revision of formatting and spelling/grammar needs to be done. 

  • Line 136-138: "‘⌊?⌋ indicates floor operation" which is the part that describes equations needs to be shifted to the contents that comes after Equation3. (as [x] notation is only used after Equation3. 
  • Line 234: "information Gain" the use of capital letter 'G' should be revised. 
  • Line 272: "Ref. [33]" reference formatting is not consistent through the paper and needs to be checked. 
  • Line 405 ~ : There are some tables describing the accuracy og Miao's method with different parameters. I think the table's content is duplicated with aforementioned figures. Either remove tables or merge them all into one (comparing result among different parameters) will better present the content.

Author Response

Responses to Reviewers' Comments on Electronics-676462

Congcong Sun, Hui Tian, Chin-Chen Chang, Yewang Chen, Yiqiao Cai,

Yongqian Du, Yonghong Chen, Chih-Cheng Chen

Reviewer #2:

Comment (2.1):

  • Line 136-138: "? indicates floor operation" which is the part that describes equations needs to be shifted to the contents that comes after Equation3. (as [x] notation is only used after Equation3.
  • Line 234: "information Gain" the use of capital letter 'G' should be revised.
  • Line 272: "Ref. [33]" reference formatting is not consistent through the paper and needs to be checked.
  • Line 405 ~ : There are some tables describing the accuracy og Miao's method with different parameters. I think the table's content is duplicated with aforementioned figures. Either remove tables or merge them all into one (comparing result among different parameters) will better present the content.

Response to Reviewer Comment (2.1):

  • Thank you for your valuable advice. We are sorry that in the previous version, Equations 1 and 2 were not correct. In fact, is used in Equations 1, 2 and 3. We have corrected the Equations 1 and 2 in this revision. The corresponding revision can be found in lines 135 and line 139.
  • We have corrected them as per your instruction, which can be found in line 234.
  • We have double-checked all the references and listed them in the required manner.
  • We have removed the duplicated tables. The corresponding revision can be found in Section 4.2.
